# Performance of Integrated Near-Infrared Spectroscopy and Intravascular Ultrasound (NIRS-IVUS) System against Quantitative Flow Ratio (QFR)

**DOI:** 10.3390/diagnostics11071148

**Published:** 2021-06-23

**Authors:** Magdalena M. Dobrolińska, Paweł M. Gąsior, Elżbieta Pociask, Grzegorz Smolka, Andrzej Ochala, Wojciech Wojakowski, Tomasz Roleder

**Affiliations:** 1Division of Cardiology and Structural Heart Diseases, Medical University of Silesia, 40-635 Katowice, Poland; magdalena.dobrolinska@gmail.com (M.M.D.); p.m.gasior@gmail.com (P.M.G.); gsmolka@me.com (G.S.); aochala1@gmail.com (A.O.); wwojakowski@sum.edu.pl (W.W.); 2Department of Biocybernetics and Biomedical Engineering, AGH University of Science and Technology, 30-059 Krakow, Poland; elzbieta.pociask@gmail.com; 3Regional Specialist Hospital, Research and Development Center, 51-124 Wroclaw, Poland

**Keywords:** quantitative flow ratio, near-infrared spectroscopy, intravascular ultrasound, coronary artery disease, ischemia

## Abstract

Quantitative flow ratio (QFR) is a new opportunity to analyze functional stenosis during invasive coronary angiography. Together with a well-known intravascular ultrasound (IVUS) and a new player in the field, near-infrared spectroscopy (NIRS), it is gaining a lot of interest. The aim of the study was to compare QFR results with integrated IVUS-NIRS results acquired simultaneously in the same coronary lesion. We retrospectively enrolled 66 patients in whom 66 coronary lesions were assessed by NIRS-IVUS and QFR. Lesions were divided into two groups based on QFR results as QFR-positive group (QFR ≤ 0.8) or QFR-negative group (QFR > 0.8). Based on ROC curve analysis, the best cut-off values of minimal lumen area (MLA), minimal lumen diameter (MLD) and percent diameter stenosis for predicting QFR ≤ 80 were 2.4 (AUC 0.733, 95%CI 0.61, 0.834), 1.6 (AUC 0.768, 95%CI 0.634, 0.872) and 59.5 (AUC 0.918, 95%CI 0.824, 0.971), respectively. In QFR-positive lesions, the maxLCBI_4mm_ was significantly higher than in QFR-negative lesions (450.12 ± 251.0 vs. 329.47 ± 191.14, *p* = 0.046). The major finding of the present study is that values of IVUS-MLA, IVUS-MLD and percent diameter stenosis show a good efficiency in predicting QFR ≤ 0.80. Moreover, QFR-positive lesions are characterized by higher maxLCBI_4mm_ as compared to the QFR-negative group.

## 1. Introduction

Fractional flow reserve (FFR), which enabled analyzing the hemodynamic significance of coronary stenosis, was a real game-changer in the diagnosis and treatment of coronary artery disease (CAD) [1,2]. Recently developed quantitative flow ratio (QFR), which computes FFR without the necessity of drug-induced hyperemia or utilization of additional pressure wire [3], is a promising technology with the potential to improve outcomes of percutaneous coronary interventions (PCI). QFR applies fluid dynamics equations and is calculated from three-dimensional quantitative coronary angiography (3D-QCA). It was previously validated and showed high diagnostic accuracy in identifying hemodynamically significant stenosis and the prediction of ≤0.8 FFR [4,5,6].

Not only the functional severity of the lesion but also morphology plays a significant role in the stenosis assessment. Intravascular ultrasound (IVUS) has proven its value in the analysis of plaque morphology, and as a result, an IVUS-guided PCI demonstrated a reduction in adverse events and cardiovascular death [7,8,9]. Besides the fact that IVUS itself is not sufficient to replace the guidance of FFR during PCI, the relationship between the functional severity of the stenosis and parameters assessed by IVUS, including minimal lumen area (MLA) and minimal lumen diameter (MLD), was also shown [10,11,12].

Importantly, if the near-infrared spectroscopy (NIRS) is added to IVUS, it enables differentiating between lipidic and fibrotic plaques [13]. NIRS detects lipids within plaques, and the amount of lipids is measured as a lipid core burden index (_max_LCBI_4mm_). Lesions with _max_LCBI_4mm_ ≥ 265 are identified as thin cap fibrous atheroma (TCFA) [14] and are associated with an increased risk of post-PCI myocardial infarction (MI) [15,16].

The aim of this study was to assess the value of parameters measured by integrated NIRS-IVUS system in the detection of significant stenosis defined by QFR.

## 2. Materials and Methods

### 2.1. Study Population

We retrospectively enrolled 66 patients diagnosed with chronic coronary syndromes (CCS) and acute coronary syndromes (ACS) between 2012 and 2015, in the high-volume tertiary center (Figure 1). Each of the enrolled patients underwent integrated NIRS-IVUS imaging and had two angiographic images acquired at different 25° angles, based on which we calculated QFR. Patients were divided into two groups based on QFR results. Those with QFR ≤ 0.8 were included in the QFR-positive group (*n* = 37), while others were included in the QFR-negative group (*n* = 29). Exclusion criteria were as follows: stent restenosis as target lesion, aorto-ostial stenosis, bifurcation lesions, tandem lesions, renal failure (creatinine >1.5 mg/dL), hemodynamic compromise and contrast allergy. None of the patients developed any complications due to integrated NIRS-IVUS imaging. Clinical demographics and medical history were obtained from hospital records. The study conformed to the Declaration of Helsinki. Due to retrospective design, further application was not needed.

### 2.2. Invasive Coronary Angiography

The PCI was performed under angiography guidance and neither integrated NIRS-IVUS system nor QFR data were used for this purpose (Figure 2a). The region to treat was selected by the operator after the diagnostic angiogram. In all of the included lesions, the drug-eluting stent (DES) was implanted. The study projections were acquired at a minimum of 12.5 frames per second with continuous and brisk contrast injections without any zooming or table movements.

### 2.3. QFR Measurement

QFR was computed with the QAngio XA-3D/QFR solution (Medis Medical Imaging Systems bv., Leiden, the Netherlands) based on a previously published method (Figure 2b) [17]. QFR was evaluated in lesions, which were previously analyzed with NIRS-IVUS. The left main artery was excluded from the analysis. The retrospective patient selection for the study was based on angiographic image projections which were acquired at different 25° angles. Images with low angiographic quality and poor contrast filling were excluded from further analysis. Only focal lesions were included. For each study, the end-diastolic frame was used for the reconstruction of the segmented vessel. After the vessel references and lesion were marked, the lumen contour was automatically delineated by validated algorithms. In the case of suboptimal angiographic image quality, manual correction was allowed. The contrast frame count was performed in an angiographic run. Frame-count-based contrast QFR was used for each analysis. In our study, the vessel contrast QFR was considered a main parameter for each analyzed coronary artery. Vessel QFR was calculated for the entire contoured segment. The QFR analysis was performed by one observer who was blind to the results of the PCI procedure and NIRS-IVUS analysis results. The percent diameter stenosis (%DS) was assessed using 2D-QCA.

### 2.4. NIRS-IVUS Analysis

The integrated NIRS-IVUS system was used to perform a culprit lesion analysis before and after stent implantation (Figure 2c,d). For this study, only a preimplantation image analysis was used. Before the insertion of the integrated NIRS-IVUS system, heparin anticoagulation (activated clotting time >300 s) was used and followed by administration of intracoronary nitroglycerine (100–200 µm). The automated pullback started with a speed of 0.5 mm/s (240 rotations/min) when the 2.4 Fr. TVC Insight Catheter (InfraReDx, TVC Imaging System, Burlington, MA, USA) was positioned at least 10 mm distal to the target lesion. The pullback stopped when the TVC catheter entered the guiding catheter. Quantitative IVUS measurements were performed in every millimeter within the region of interest (ROI), which had to be at least 4 mm long. We analyzed IVUS parameters measured on cross-sectional IVUS images, including minimal lumen area (MLA), minimal lumen diameter (MLD), lesion length and plaque burden. Plaque burden was calculated as total plaque area divided by EEM CSA × 100 (%). The remodelling index (RI) was calculated by dividing EEM area at the MLA by the reference EEM area. Negative and positive remodeling were defined as RI ≤ 0.95 and RI ≥ 1.05, respectively. RI between these values was defined as a nonremodeled vessel.

The chemical composition of the plaque within ROI was analyzed using NIRS. On NIRS chemogram, 1 pixel every 0.1 mm on the x-axis displays the pullback position, while 1 pixel every 1° shows the circumferential position. The fraction of yellow pixels within the ROI, indicating lipids, was calculated as a lipid core burden index (LCBI). Within the ROI, the maximal amount of lipids in 4 mm was automatically chosen by the software and expressed as maximal LCBI in 4 mm (maxLCBI_4mm_). Thin cap fibrous atheroma (TCFA) suspected lesions were defined as maxLCBI_4mm_ ≥ 265. NIRS-IVUS data were analyzed off-line using CAAS intravascular software (Pie Medical Imaging BV, Maastricht, the Netherlands).

### 2.5. Statistical Analysis

Continuous variables are presented as means with standard deviation (±SD) or medians with interquartile intervals (IQR, 1st, 3rd). Categorical data are shown as the number or percentage (%). For the comparison, the one-way ANOVA and Mann–Whitney test were used [18]. Correlation was measured using Pearson’s or Spearman’s rank-order correlation. The categorical data were compared using Fischer’s exact test or chi-square test. Receiver operating characteristic (ROC) curve analyses were performed to identify the optimal cut-off values of IVUS parameters for the prediction of hemodynamic significance with maximum accuracy [19]. A value of *p* < 0.05 was considered statistically significant. MedCalc version 15.8 (MedCalc 15.8, MedCalc Software, Ostend, Belgium) and SPSS (SPSS v.23, Armonk, NY, USA) were used for statistical analysis.

## 3. Results

### 3.1. Patients Characteristics

Patients’ characteristics are summarized in Table 1. We analyzed 66 focal lesions in 66 patients. Patients from the QFR-positive group were not significantly younger than patients from the QFR-negative group (63.3 ± 10.34 vs. 62.8 ± 10.9; *p* = 0.767). There were no significant differences in the percentage of patients who had hypertension (43.2% vs. 79.3%, *p* = 0.246), dyslipidemia (35.1% vs. 62.0%, *p* = 0.114) or diabetes mellitus (21.6% vs. 20.7%, *p* = 0.246) in QFR-positive and QFR-negative groups. The QFR-negative group was characterized by a higher percentage of patients who had PCI in the past (34.5% vs. 10.8%, *p* = 0.016).

### 3.2. Lesion Characteristics

Of the lesions analyzed, 46.9% were located in the left anterior descending coronary artery (LAD). The mean diameter of the proximal reference was equal to 2.89 ± 0.50 mm. The mean percent diameter stenosis was 63.25 ± 14.93%. The median MLA and MLD were 2.35 (1.97, 2.95) and 1.5 (1.5, 1.7), respectively. The median vessel contrast QFR value for a total of 66 lesions was equal to 0.8 (0.7, 0.9). Lesions with QFR ≤ 0.8 were included in the QFR-positive group (*n* = 37), while others were included in the QFR-negative group (*n* = 29). Based on ROC curve analysis, the best cut-off values of MLA, MLD and percent diameter stenosis for predicting QFR ≤ 0.80 were 2.4 (AUC 0.733, 95%CI 0.61, 0.834, *p* < 0.001), 1.6 (AUC 0.768, 95%CI 0.634, 0.872, *p* < 0.001) and 59.5 (AUC 0.918, 95%CI 0.824, 0.971, *p* < 0.001), respectively. ROC curves are displayed in Figure 3a–c.

### 3.3. QFR and NIRS-IVUS Lesion Analysis

The mean vessel QFR in positive and negative groups was equal to 0.67 ± 0.14 and 0.9 ± 0.06, respectively. In the QFR-positive group, only 59.5% of lesions had diameter stenosis >70% measured by 2D-QCA. NIRS-IVUS results are reported in Table 2. In the QFR-positive group, lesions were not significantly longer than QFR-negative lesions (27.7 ± 10.74 vs. 22.91 ± 11.02, *p* = 0.48). The QFR-positive group was characterized by smaller MLA (2.2 ± 0.42 vs. 3.12 ± 1.44; *p* = 0.007) and MLD (1.5 (1.5, 1.6) vs. 1.7 (1.5, 1.9); *p* = 0.001) as compared to QFR-negative group. In QFR-positive lesions, the maxLCBI_4mm_ was significantly higher than in QFR-negative lesions (450.12 ± 251.0 vs. 329.47 ± 191.14, *p* = 0.046); however, there was no difference in the amount of TCFA lesions between the two groups. There were no significant differences in plaque volume and plaque burden between the two groups (respectively *p* = 0.252, *p* = 0.286). There was also no difference in lumen volume and EEM volume between both groups (respectively *p* = 0.737, *p* = 0.658). We did not find a difference in RI between both groups as well (1.02 (0.8, 1.27) vs. 1.00 (0.84, 1.44); *p* = 0.69).

### 3.4. QFR and NIRS-IVUS Correlation Analysis

The percent diameter stenosis in the QFR-positive and QFR-negative groups was 71.98 ± 8.67% and 49.42 ± 12.13% (*p* = 0.000), respectively. Correlations are displayed in Table 3. In the QFR-positive group there was no correlation between vessel QFR and MLA (*r* = 0.151, *p* = 0.721), MLD (*r* = 0.064, *p* = 0.722), plaque volume (*r* = −0.229, *p* = 0.173) or plaque burden (*r* = −0.227, *p* = 0.176). In the QFR-negative group, there was also no correlation between vessel QFR and MLA (*r* = 0.151, *p* = 0.435), MLD (*r* = 0.388, *p* = 0.082), plaque volume (*r* = −0.212, *p* = 0.271) or plaque burden (*r* = −0.177, *p* = 0.358).

## 4. Discussion

The main finding of this study is that, based on NIRS-IVUS analysis, MLA and MLD were better than plaque burden as predictors of hemodynamically significant stenosis based on QFR of ≤0.80, and %DS measured in 2D-QCA was also a better predictor than plaque burden. Moreover, the QFR-positive group was characterized by higher maxLCBI4mm. Interestingly, we did not find any correlation between vessel QFR and MLA, MLD, plaque volume or plaque burden in the QFR-positive group or the QFR-negative group.

Currently, the invasively measured FFR remains the gold standard in the assessment of coronary stenosis severity. Despite its undeniable clinical value, the costs of pressure wires and the need for hyperemia limit its everyday use in many centers. Importantly, the response to adenosine-induced hyperemia not only varies between individuals, including increased heart rate or decreased blood pressure, but also causes patient discomfort [20,21,22]. Even though adenosine-free methods, including instantaneous wave-free ratio (iFR) or resting full-cycle ratio (RFR), are available, the use of additional wire is still required. Owing to the availability of QFR, which is an angiography technique computing FFR without drug-induced hyperemia or a pressure wire, it has gained a lot of interest.

FAVOR Pilot and FAVOR II were the first studies demonstrating that wire-free QFR is superior to standard quantitative coronary angiography for evaluation of intermediary coronary artery stenosis [23]. When the QFR was applied to the 2D-QCA, the diagnostic accuracy of combined QFR-QCA increased to 92.7% as compared to quantitative coronary angiography [4]. Furthermore, the functional coronary lesion assessment by QFR showed a good diagnostic accuracy as compared to FFR, which indicates that it is a reliable method for the assessment of coronary hemodynamics [24,25,26]. Recently, QFR was successfully used to select the most appropriate patients for further FFR evaluation [5]. There was also a high agreement between QFR and iFR [27,28]. Moreover, QFR measurement has a low interobserver variability, which proves its value as a feasible method in everyday practice [29]. Additionally, the results of the WIFI-II study showed that the addition of QFR into everyday assessment may reduce the use of pressure wires [25]. In our study, based on 2D-QCA, all of 66 lesions underwent stent implantation. According to the QFR results, only 56% of them were hemodynamically significant. Therefore, based on QFR results, 44% of lesions treated in our study might have been potentially safely deferred.

During the last two decades, many attempts have been made to assess coronary hemodynamics based on lesion anatomy [30]. According to Okabe et al., IVUS is considered a valuable tool to guide PCI which not only assesses the morphology of the lesion but also reduces the occurrence of stent thrombosis [31]. As the insertion of two additional wires during one procedure is not considered a desirable approach, the ability of IVUS to assess functional lesion severity attracts attention. Although several studies, including large international multicenter trials, have investigated the relationship between FFR severity and IVUS parameters, there are no data comparing IVUS to QFR.

As reported by Voros et al. [32], within the IVUS-derived measurements, MLD had the strongest correlation with FFR, whereas MLA was the best predictor by ROC and multivariable analysis. The FIRST study showed a moderate correlation of IVUS results with the FFR values. However, the cut-off values of MLA measured by IVUS for detecting hemodynamically relevant stenosis (<2.4, <2.7 and <3.6 mm^2^) dependent on vessel size (reference vessel diameters <3.0, 3.0–3.5 and >3.5 mm, respectively) were found [10]. Specifically, IVUS-derived MLA ≥ 2.4 mm^2^ was considered useful to rule out lesions with FFR < 0.80 [33]. Furthermore, a meta-analysis showed that for lesions with an angiographic diameter greater than 3 mm, the MLA cut-off was equal to 2.8 mm^2^, while for lesions smaller than 3 mm, the MLA cut-off was 2.4 mm^2^ [34]. The cut-off MLA value from our study, which was calculated based on QFR analysis, is comparable to results from other studies when FFR as a reference was used. Moreover, Gonzalo et al. found a cut-off value of IVUS-derived MLD equal to 1.59 mm with FFR as a reference, while the IVUS-MLD cut-off value in our study equals 1.6 mm [35]. As compared to the analyses in which FFR was used as a reference, our results with QFR as a reference are similar. Interestingly, there was no correlation between anatomical values, including MLA, MLD, plaque burden and plaque volume, and functional vessel QFR, even though both MLA and MLD were significantly lower in the QFR-positive group.

As far as morphology is concerned, NIRS enables not only the periprocedural analysis of chemical plaque composition but also the quantitative measurement of lipid content as a maxLCBI4mm [36,37]. It should be noted that neither QFR nor IVUS enable the determination of lipid-rich lesions. NIRS simultaneously distinguishes lipid-core plaques (LCP), which are associated with increased risk of periprocedural MI and restenosis rates following stent implantation [38,39,40]. The specific lipid-rich lesions with maxLCBI4mm ≥265 were defined as TCFA [14]. In our study, QFR-positive lesions were characterized by greater maxLCBI4mm. Importantly, in our study, there was no significant difference in the number of TCFA lesions between QFR-positive and QFR-negative groups. As we know from the PROSPECT-II study, a lipid-rich lesion increases the risk of coronary events in the future [13]. According to that, the integrated NIRS-IVUS system may not only detect significant stenosis, but also enable planning further lipid-lowering therapy and follow-up of patients with hemodynamically insignificant lesions.

Initial studies on QFR indicate its great potential due to wire-free, nonhyperemic measurement and the fact that the addition of QFR to the standard procedure does not increase the contrast use. Furthermore, the hybrid NIRS-IVUS-QFR guidance enables not only the functional but also anatomical and morphological plaque assessment. Moreover, the adjustment of NIRS to IVUS gives a unique opportunity to plan further treatment and optimize the lipid-lowering therapy in patients with TCFA lesions.

This study has several limitations. First, this study was a single-center, retrospective study involving a small number of patients. Second, we selected only lesions that were evaluated by NIRS-IVUS and met the criteria for QFR measurement, which may result in selection bias. Additionally, lesions were only analyzed after angioplasty followed by stent implantation, which may have led to an inaccurate assessment of the association between QFR and IVUS. Moreover, we did not use FFR, which is considered a gold standard. A larger and prospective study is needed to verify the results of this study.

## 5. Conclusions

In the present study, we found that IVUS values of IVUS-MLA, IVUS-MLD and percent diameter stenosis show good accuracy in predicting QFR ≤ 80. Additionally, we showed that QFR-positive lesions are characterized by higher maxLCBI_4mm_ as compared to the QFR-negative group. However, we did not find any correlation between vessel QFR and MLA, MLD, plaque volume or plaque burden in the QFR-positive group or the QFR-negative group. As the assessment of stenosis function with a pressure wire has some limitations due to contraindication for adenosine infusion or patient discomfort during hyperemia, the combination of NIRS-IVUS and QFR may play an important role in further diagnosis and treatment.

## Figures and Tables

**Figure 1 diagnostics-11-01148-f001:**
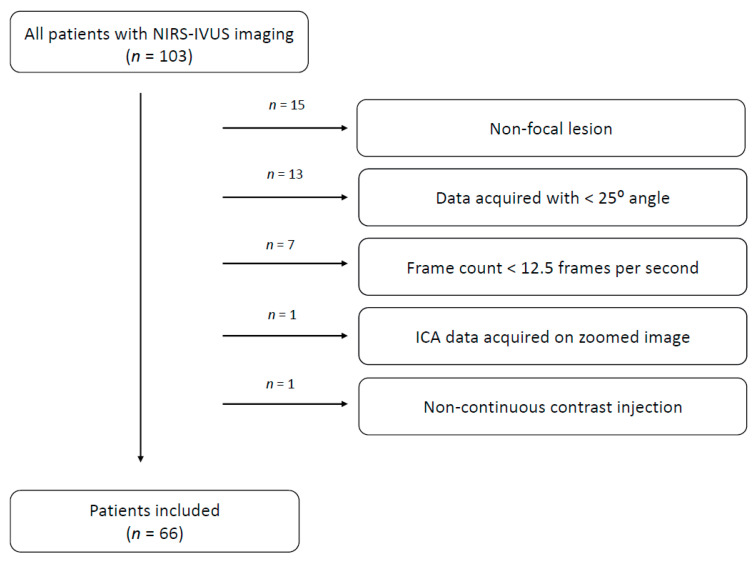
Flowchart of patient selection.

**Figure 2 diagnostics-11-01148-f002:**
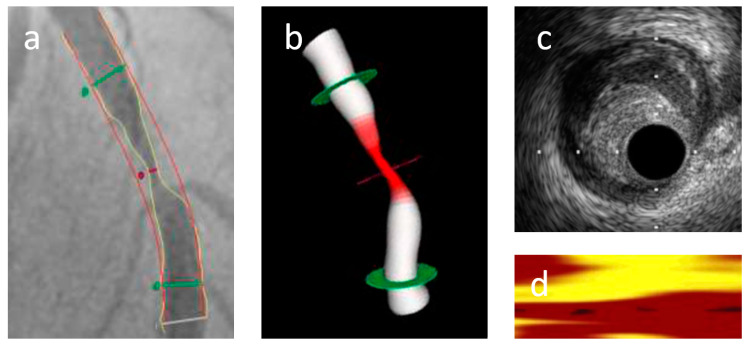
Representative of QFR-NIRS-IVUS imaging before stent implantation: (**a**) baseline angiography, (**b**) QFR, (**c**) IVUS and (**d**) NIRS analysis of a patient with stenosis of the LAD.

**Figure 3 diagnostics-11-01148-f003:**
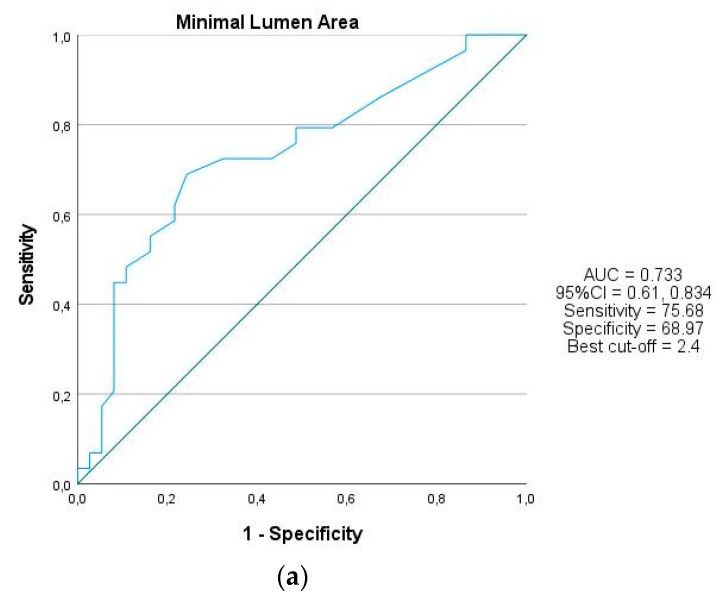
ROC curves of anatomical parameters predicting QFR ≤ 0.8. (**a**) The ROC curve for percent diameter stenosis derived from angiography. Best cut-off 59.5%; (**b**) The ROC curve for MLD derived from IVUS. Best cut-off 1.6 mm; (**c**) The ROC curve for MLA derived from IVUS. Best cut-off 2.4 mm^2^.

**Table 1 diagnostics-11-01148-t001:** Patient characteristics.

	QFR-Positive *n* = 37	QFR-Negative *n* = 29	*p*
**Clinical Demographics**			
Age (years)	63.3 ± 10.34	62.8 ± 10.9	0.767
Body mass index kg/m^2^	21.89 ± 12.87	18.7 ± 12.3	0.538
Prior MI n%	4 (10.8)	7 (24.1)	0.833
Prior PCI n%	4 (10.8)	10 (34.5)	0.016
Prior CABG n%	0 (0)	0 (0.0)	
Dyslipidemia n%	13 (35.1)	18 (62.0)	0.114
Hypertension n%	16 (43.2)	23 (79.3)	0.246
Diabetes mellitus n%	8 (21.6)	6 (20.7)	0.925
TCH	154.3 ± 36.9	128.0 ± 45.0	0.093
LDL	71.8 ± 38.4	80.0 ± 28.7	0.511
HDL	41.46 ± 13.52	31.4 ± 19.9	0.120
TG	124.5 (77.75, 145.25)	76 (48.0, 136.0)	0.656
GFR	61.6 ± 23.15	57.8 ± 27.47	0.681

Table 1: Variables are displayed as mean ± SD when a normal distribution is present, or as median (1st–3rd quartile) when a normal distribution is not present. For each variable, the percentage of patients involved (*n*%) is given.

**Table 2 diagnostics-11-01148-t002:** NIRS, IVUS and QFR characteristics.

	QFR-Positive	QFR-Negative	*p*
**Indication for Coronary Angiography**			
ACS	10 (27.0)	6 (20.7)	0.771
**Lesion Location**			
LAD	19 (51.3)	12 (41.4)	
Cx	11 (29.7)	7 (24.1)	
RCA	7 (19.0)	10 (34.5)	
**QFR Analysis**			
Diameter stenosis	71.98 ± 8.67	49.42 ± 12.13	0.000
Vessel QFR contrast	0.67 ± 0.14	0.9 ± 0.06	0.000
**IVUS Analysis**			
Stenosis length	27.7 ± 10.74	22.91 ± 11.02	0.480
Lumen volume	126.97 ± 68.51	133.93 ± 81.56	0.737
EEM volume	353.74 ± 207.49	327.2 ± 222.5	0.658
EEM area at MLA	10.59 ± 3.4	13.04 ± 5.5	0.048
Plaque volume	226.16 ± 155.57	193.27 ± 149.16	0.252
Plaque burden	77.2 ± 7.82	75.01 ± 7.43	0.286
Minimal lumen area	2.2 ± 0.42	3.12 ± 1.44	0.007
Minimal lumen diameter	1.5 (1.5, 1.6)	1.7 (1.5, 1.9)	0.001
Total plaque area	8.3 (5.45, 10.55)	8.4 (7.5, 10.85)	0.347
RI	1.02 (0.8, 1.27)	1.00 (0.84, 1.44)	0.690
MLA < 2.4 mm^2^	28 (75.67)	9 (31.0)	0.000
**NIRS Analysis**			
maxLCBI_4mm_	450.12 ± 251.0	329.47 ± 191.14	0.046
TCFA lesions	22 (59.45)	21 (72.41)	0.276

Table 2: Variables are displayed as mean ± SD when a normal distribution is present or as median (1st–3rd quartile) when there a normal distribution is not present. For each variable, the percentage of patients involved (*n*%) is given. ACS—acute coronary syndrome; Cx—circumflex coronary artery; EEM—external elastic membrane; LAD—left anterior descending; maxLCBI_4mm_—maximal lipid core burden index in 4 mm; MLA—minimal lumen area; RI—remodeling index; RCA—right coronary artery; TCFA—predicted thin cap fibrous atheroma by NIRS.

**Table 3 diagnostics-11-01148-t003:** The correlations between MLA, PB, MaxLCBI_4mm_, PV, MLD and vessel contrast QFR.

	QFR-Positive		QFR-Negative	
	Vessel QFR (r)	*p*	Vessel QFR (*r*)	*p*
MLA	0.151	0.721	0.151	0.435
PB	−0.227	0.176	−0.177	0.358
maxLCBI_4mm_	−0.198	0.239	−0.033	0.866
PV	−0.229	0.173	−0.212	0.271
MLD	0.064	0.722	0.388	0.082

Table 3: maxLCBI_4mm_—maximal lipid core burden index in 4 mm; MLA—minimal lumen area; MLD—minimal lumen diameter; PB—plaque burden; PV—plaque volume.

## Data Availability

Data sharing is not applicable to this article.

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
