# Peer review of "Performance of Integrated Near-Infrared Spectroscopy and Intravascular Ultrasound (NIRS-IVUS) System against Quantitative Flow Ratio (QFR)"

_diagnostics, 2021, doi:10.3390/diagnostics11071148_

Round 1
Reviewer 1 Report
Dear All,
The authors compared QFR results with integrated IVUS-NIRS analysis. Presented study had very small number of patients and provided data was obtained from single-centre. The manuscript is well written, with potentially interesting topic. Despite limitations of this study, presented manuscript might be appropriate for the Journal, but a few more issues should be considered before acceptance:
- Was professional corelabe involved in evaluation of QFR? How many specialists evaluated measurements? Impact of interobserver variability?
- What is clinical importance of this study? What is potential clinical utility of these results? Could this data improve everyday clinical practise and patients care?
Author Response
Dear Reviewer,
Thank you for your valuable comments. Please see our answers below.
The authors compared QFR results with integrated IVUS-NIRS analysis. Presented study had very small number of patients and provided data was obtained from single-centre. The manuscript is well written, with potentially interesting topic. Despite limitations of this study, presented manuscript might be appropriate for the Journal, but a few more issues should be considered before acceptance:
- Was professional corelabe involved in evaluation of QFR? How many specialists evaluated measurements? Impact of interobserver variability?
QFR data was not sent to the official QFR core lab. QFR analysis was performed by trained operator blinded to the results of 2D-QCA and NIRS-IVUS.
- What is clinical importance of this study? What is potential clinical utility of these results? Could this data improve everyday clinical practise and patients care?
The most important clinical implication is that when the combination of IVUS-NIRS and QFR is used for stenosis assessment, the adenosine is no longer needed, which might potentially be useful for patients with contraindications to adenosine stress test. However, larger, and prospective study is needed to verify the results of this study.
Reviewer 2 Report
The authors have carried out an interesting study in which they compared the relationships between NIRS/IVUS and the QFR in a population of 66 patients; this was a retrospective study. The value of the NIRS component is of course that NIRS provides information about the lipid content in the plaque. They found that the IVUS values of obstruction area, diameter and % diameter stenosis demonstrate a high accuracy in predicting a flow limiting stenosis based on a QFR < 0.8. Also, QFR positive lesions exhibited a higher lipid content than QFR-negative lesions; however, they did not find a sufficient correlation between the lipid content parameter and the QFR.
On the other hand, the lipid content provides additional information in both the QFR positive and negative groups and that most likely will lead to different treatment strategies, certainly in the QFR positive group. The fact that both anatomy by IVUS and QFR, coronary physiology by QFR, as well as lipid content by NIRS analysis are available, makes this a rather unique study.
Author Response
Dear Reviewer,
Thank you so much for your valuable comment. We really appreciate that you found our manuscript interesting.
The authors have carried out an interesting study in which they compared the relationships between NIRS/IVUS and the QFR in a population of 66 patients; this was a retrospective study. The value of the NIRS component is of course that NIRS provides information about the lipid content in the plaque. They found that the IVUS values of obstruction area, diameter and % diameter stenosis demonstrate a high accuracy in predicting a flow limiting stenosis based on a QFR < 0.8. Also, QFR positive lesions exhibited a higher lipid content than QFR-negative lesions; however, they did not find a sufficient correlation between the lipid content parameter and the QFR.
On the other hand, the lipid content provides additional information in both the QFR positive and negative groups and that most likely will lead to different treatment strategies, certainly in the QFR positive group. The fact that both anatomy by IVUS and QFR, coronary physiology by QFR, as well as lipid content by NIRS analysis are available, makes this a rather unique study.
Reviewer 3 Report
The authors aimed to compare QFR results with integrated IVUS-NIRS results acquired simultaneously in the same coronary lesion. They enrolled 66 coronary lesions assessed with both modalities. They found that values of IVUS-MLA (2,4mm2), IVUS- MLD (1,6mm) and %Diameter Stenosis (59,5%) showed a good efficiency in predicting QFR≤0.80. Moreover, QFR-positive lesions characterize higher maxLCBI4mm as compared to QFR-negative group.
My remarks:
- QFR showed a high diagnostic accuracy in identifying hemodynamically significant stenosis, and the prediction of ≤ 0.8 FFR but also iFR ≤0.89, please revise (eg. Adv Med Sci. 2021 Mar;66(1):1-5).
- How the patients were chosen? Were they the only ones who had NIRS performed? 12.5 fps was not a routine approach back in 2012, as QFR was introduced in 2014. Please provide a study flowchart showing how many pts dropped out from QFR assessment due tovarious reasons (overlap, low fps, tortuosity etc).
- Were routine coronary physiology performed for intermediate lesions?
- Did the authors perform NIRS and QFR assessment after revascularization in QFR <0.80 group? This could provide also interesting results.
Author Response
Dear Reviewer,
Thank you for your valuable comments. Please see our answers below.
The authors aimed to compare QFR results with integrated IVUS-NIRS results acquired simultaneously in the same coronary lesion. They enrolled 66 coronary lesions assessed with both modalities. They found that values of IVUS-MLA (2,4mm2), IVUS- MLD (1,6mm) and %Diameter Stenosis (59,5%) showed a good efficiency in predicting QFR≤0.80. Moreover, QFR-positive lesions characterize higher maxLCBI4mm as compared to QFR-negative group.
My remarks:
- QFR showed a high diagnostic accuracy in identifying hemodynamically significant stenosis, and the prediction of ≤ 0.8 FFR but also iFR ≤0.89, please revise (eg. Adv Med Sci. 2021 Mar;66(1):1-5).
Thank you for this comment, we added a short comment in the discussion.
- How the patients were chosen? Were they the only ones who had NIRS performed? 12.5 fps was not a routine approach back in 2012, as QFR was introduced in 2014. Please provide a study flowchart showing how many pts dropped out from QFR assessment due tovarious reasons (overlap, low fps, tortuosity etc).
Thank you for this comment. We took into consideration only those patients, who had a NIRS-IVUS imaging. Additionally, as mentioned in methods section, only images acquired with 12.5 fps were included. Therefore, not all patients with NIRS-IVUS analysis could have been included in the study. We prepared a flowchart (Figure 1) of patients’ selection.
- Were routine coronary physiology performed for intermediate lesions?
No, we did not perform routine physiology assessment in intermediate lesions. We only included angioplasty with stent implantation based on angiography results.
- Did the authors perform NIRS and QFR assessment after revascularization in QFR <0.80 group? This could provide also interesting results.
Thank you for this comment. We do agree that this analysis would be very interesting and informative. However, we could not perform analysis after stent implantation due to the lack of angiographic image projections which were acquired at different 25⁰ angles.
Reviewer 4 Report
Dobrolińska and colleagues investigated the relationship between QFR and NIRS-IVUS. The idea is absolutely interesting but the small number of patients (66) divided into 2 groups based on the QFR result make the analysis inconclusive and not worthy of publication.
Author Response
Dobrolińska and colleagues investigated the relationship between QFR and NIRS-IVUS. The idea is absolutely interesting but the small number of patients (66) divided into 2 groups based on the QFR result make the analysis inconclusive and not worthy of publication.
Dear Reviewer, we added a limitations section at the end of the discussion.
Reviewer 5 Report
Authors proposed interesting combined NIRS and IVUS imaging modality and showed some reasonable quantitative flow ratio around 0.8. The article shows the meaningful parameter data value for integrated NIRS-IVUS system. However, Figures quality for experimental data need to be improved. There are some missing references. Therefore, the manuscript could be minor revision if authors follow the suggestions.
1. Please provide Acknowledgments and data availability sections.
2. Please reduce spaces for journal names in the reference sections.
3. Please used reduced journal names in the reference sections.
4. Please increase label sizes of Figure 2a and b because quality looks so bad.
5. Please reduce Figure 2 labels space.
6.Please provide the reference for the sentence (Intravascular ultrasound (IVUS) has proven its value in the analysis of plaque morphology ) with the reference (De Korte, Chris L., et al. "Characterization of plaque components with intravascular ultrasound elastography in human femoral and coronary arteries in vitro." Circulation 102.6 (2000): 617-623. ) or another reference.
7.Please provide the reference for the sentence (For the comparison the one-way ANOVA and Mann-Whitney-test were used ) with the reference (Choi, H.; Choe, S.-w. Therapeutic Effect Enhancement by Dual-Bias High-Voltage Circuit of Transmit Amplifier for Immersion Ultrasound Transducer Applications. Sensors 2018, 18, 4210).
8.Please provide the reference for the sentence (Receiver operating characteristic (ROC) curve analyses were performed to identify the ) with the reference (Mirmiran, P. A. R. V. I. N., A. Esmaillzadeh, and F. Azizi. "Detection of cardiovascular risk factors by anthropometric measures in Tehranian adults: receiver operating characteristic (ROC) curve analysis." European journal of clinical nutrition 58.8 (2004): 1110-1118. ) or another reference.
9.Please reduce space for each paragraph in the manuscript. Please check MDPI format.
10. Please do not use underbar in Line 33.
Author Response
Dear Reviewer,
Thank you for your valuable comments. Please see our answers below.
Authors proposed interesting combined NIRS and IVUS imaging modality and showed some reasonable quantitative flow ratio around 0.8. The article shows the meaningful parameter data value for integrated NIRS-IVUS system. However, Figures quality for experimental data need to be improved. There are some missing references. Therefore, the manuscript could be minor revision if authors follow the suggestions.
- Please provide Acknowledgments and data availability sections.
We added these sections.
Please reduce spaces for journal names in the reference sections.
We reduced the spaces between journal names.
Please used reduced journal names in the reference sections.
Names of following four journals: Lancet, EuroIntervention, Circulation, and Sensors, remain as full names. However, it seems to us that there is no official short name for these journals.
Please increase label sizes of Figure 2a and b because quality looks so bad.
The label size is increased. We also attached high resolution images as a zip file in the previous submission. Is that possible to use the high resolution images which we added earlier?
Please reduce Figure 2 labels space.
We decreased the label space.
6.Please provide the reference for the sentence (Intravascular ultrasound (IVUS) has proven its value in the analysis of plaque morphology ) with the reference (De Korte, Chris L., et al. "Characterization of plaque components with intravascular ultrasound elastography in human femoral and coronary arteries in vitro." Circulation 102.6 (2000): 617-623. ) or another reference.
7.Please provide the reference for the sentence (For the comparison the one-way ANOVA and Mann-Whitney-test were used ) with the reference (Choi, H.; Choe, S.-w. Therapeutic Effect Enhancement by Dual-Bias High-Voltage Circuit of Transmit Amplifier for Immersion Ultrasound Transducer Applications. Sensors 2018, 18, 4210).
8.Please provide the reference for the sentence (Receiver operating characteristic (ROC) curve analyses were performed to identify the ) with the reference (Mirmiran, P. A. R. V. I. N., A. Esmaillzadeh, and F. Azizi. "Detection of cardiovascular risk factors by anthropometric measures in Tehranian adults: receiver operating characteristic (ROC) curve analysis." European journal of clinical nutrition 58.8 (2004): 1110-1118. ) or another reference.
All three citations are provided.
9.Please reduce space for each paragraph in the manuscript. Please check MDPI format.
We reduced space for each paragraph.
Please do not use underbar in Line 33.
Thank you for this remark, this is corrected.
Round 2
Reviewer 1 Report
no further comments